# Mortality Prediction of Patients with Subarachnoid Hemorrhage Using a Deep Learning Model Based on an Initial Brain CT Scan

**DOI:** 10.3390/brainsci14010010

**Published:** 2023-12-22

**Authors:** Sergio García-García, Santiago Cepeda, Dominik Müller, Alejandra Mosteiro, Ramón Torné, Silvia Agudo, Natalia de la Torre, Ignacio Arrese, Rosario Sarabia

**Affiliations:** 1Neurosurgery Department, Rio Hortega University Hospital, 47012 Valladolid, Spain; scepedac@saludcastillayleon.es (S.C.); silvia.agudo@hotmail.es (S.A.); natalia.delatorregeijo@live.com (N.d.l.T.); iarreser@saludcastillayleon.es (I.A.); rsarabia@saludcastillayleon.es (R.S.); 2IT-Infrastructure for Translational Medical Research, University of Augsburg, 86159 Augsburg, Germany; dominik.mueller@informatik.uni-augsburg.de; 3Neurosurgery Department, Hospital Clinic de Barcelona, 08036 Barcelona, Spain; mosteiro@clinic.cat (A.M.); ramtorne@me.com (R.T.)

**Keywords:** subarachnoid hemorrhage, convolutional neural networks, artificial intelligence, mortality, prognosis, CT scan

## Abstract

Background: Subarachnoid hemorrhage (SAH) entails high morbidity and mortality rates. Convolutional neural networks (CNN) are capable of generating highly accurate predictions from imaging data. Our objective was to predict mortality in SAH patients by processing initial CT scans using a CNN-based algorithm. Methods: We conducted a retrospective multicentric study of a consecutive cohort of patients with SAH. Demographic, clinical and radiological variables were analyzed. Preprocessed baseline CT scan images were used as the input for training using the AUCMEDI framework. Our model’s architecture leveraged a DenseNet121 structure, employing transfer learning principles. The output variable was mortality in the first three months. Results: Images from 219 patients were processed; 175 for training and validation and 44 for the model’s evaluation. Of the patients, 52% (115/219) were female and the median age was 58 (SD = 13.06) years. In total, 18.5% (39/219) had idiopathic SAH. The mortality rate was 28.5% (63/219). The model showed good accuracy at predicting mortality in SAH patients when exclusively using the images of the initial CT scan (accuracy = 74%, F1 = 75% and AUC = 82%). Conclusion: Modern image processing techniques based on AI and CNN make it possible to predict mortality in SAH patients with high accuracy using CT scan images as the only input. These models might be optimized by including more data and patients, resulting in better training, development and performance on tasks that are beyond the skills of conventional clinical knowledge.

## 1. Introduction

Subarachnoid hemorrhage (SAH) is a devastating form of hemorrhagic stroke with an incidence of 6–8 persons per 100,000 inhabitants per year and a higher incidence in specific regions such as Japan, Finland or Indiana [1]. Around 70–80% of spontaneous SAHs are caused by the rupture of an intracranial aneurysm, known as aneurysmal SAH (aSAH) [2]. Despite its low incidence, aSAH is a major burden for healthcare systems due to its high mortality and morbidity rates despite optimal treatment [3].

In a modern series, 30-day mortality rates range between 27% and 44%. Little improvement has been achieved in the last decade despite extensive efforts to treat its causes or understand the pathophysiology of the many and treacherous complications that may arise along its course [4,5,6]. Predictors of in-hospital mortality include the admission clinical grade, rebleeding, delayed cerebral ischemia, treatment-related ischemia and intraventricular hemorrhage [4,7]. Early brain injury due to the initial hemorrhagic insult and aneurysm rebleeding account for most fatalities [4]. Therefore, efforts have been addressed to prevent aSAH by controlling vascular risk factors and to prevent rebleeding by granting an early exclusion of the aneurysm. The latter is the epitome of medical debate; optimal timing and the best therapeutic approach are fiercely discussed [8,9,10]. However, survival and functional results have scarcely improved during these first decades of the century. The accurate prediction of outcomes in patients with moderate to poor grades remains a challenge.

Accurate predictions in the medical field often require a large amount of data from large cohorts of patients. Although patient data are increasingly accessible, managing such complex information has led to the development of modern predictive algorithms and models based on artificial intelligence (AI). Convolutional neural networks (CNNs), a form of deep learning (DL), mimic the entangled and complex system of connections existing in biological neural structures. Nodes are organized into layers and are interconnected to generate and spread output signals resulting from multiple interlinked activation functions. CNNs can modify their behavior as they learn from their training. In addition, CNNs might consider features or variables otherwise ignored by the observer. CNNs have shown excellent performance in accurately predicting various targeted variables in the medical field based on different imaging modalities [11,12]. Some known risk factors for in-hospital mortality associated with aSAH can be identified from the initial CT scan (blood amount, intraventricular hemorrhage, edema, ischemic changes, etc.) [4,7,13,14]. This is advantageous as a single sequence of images acquired upon admission can provide most of the relevant information necessary to predict a patient’s course.

In this clinical investigation, we sought to design, create and evaluate a model based on a CNN applied to initial CT scans to predict the mortality of patients admitted to the hospital with a SAH at three months.

## 2. Materials and Methods

The present investigation was conducted following the Strengthening the Reporting of Observational Studies in Epidemiology (STROBE) [15] and the Checklist for Artificial Intelligence in Medical Imaging (CLAIM) [16] guidelines. The study protocol was approved by the Institutional Review Board (22-PI180).

### 2.1. Study Population

This was a retrospective, non-interventional study. The clinical records of a consecutive cohort of patients diagnosed with SAH admitted to our institution (Hospital Universitario Rio Hortega, Valladolid, Spain) between 2011 and 2022 were retrospectively reviewed. Additionally, a consecutive series of patients from another institution (Hospital Clinic de Barcelona, Barcelona, Spain) with the same inclusion criteria was included to test the robustness of the algorithm. Therefore, the inclusion criteria comprised aneurysmal and non-aneurysmal (perimesencephalic) spontaneous SAH diagnoses based on compatible clinical signs and a positive CT scan as well as known survival status at three months. Aneurysmal and non-aneurysmal etiologies were, respectively, established by a positive or negative AngioCT and/or digital subtraction angiography. Patients whose CT scans were acquired later than 24 h from the onset of symptoms or those that could not properly be processed were excluded.

### 2.2. Variables

Demographic data such as age, sex, cardiovascular risk factors (smoking, hypertension, diabetes, dyslipidemia and family history of SAH), the admission clinical severity scales (World Federation of Neurosurgical Societies (WFNS) and Hunt and Hess (HH) grading scales), modified Fisher (mF) scale and mortality at 3 months were obtained from the clinical records of included patients [17,18,19].

### 2.3. Image Acquisition

CT scans from the institutional and external cohorts were, respectively, acquired using Phillips Ingenuity CT (Koninklijke, The Netherlands) and Siemens Somaton CT (Munich, Germany) scanners (Appendix A).

### 2.4. Image Preprocessing

CT images were sourced using the Digital Imaging and Communications in Medicine (DICOM) format. An initial step involved transformation into the Neuroimaging Informatics Technology Initiative (NIfTI) format and employing the dicom2niix tool v1.0.20220720 (https://github.com/rordenlab/dcm2niix/releases/tag/v1.0.20220720 accessed on 18 January 2023).

To avoid negative Hounsfield units (Hus), we implemented an intensity normalization through a lossless transformation to Cormack units using the Clinical Toolbox for SPM (https://github.com/neurolabusc/Clinical accessed on 20 January 2023).

Subsequently, we conducted a brain extraction procedure using the Brain Extraction Tool (BET) from the Functional MRI of the Brain Software Library (FSL) v6.0 (https://fsl.fmrib.ox.ac.uk/fsl/fslwiki/BET/UserGuide accessed on 20 January 2023). The final step involved registration to a CT template image with dimensions of 1 × 1 × 1 mm and a 193, 229 and 193 size by applying diffeomorphic registrations using symmetric normalization (SyN) from the Advanced Normalization Tools program (https://github.com/ANTsX/ANTs accessed on 21 January 2023).

### 2.5. Neural Network

In our research, the AUCMEDI (Automated Classification of Medical Images) framework (https://frankkramer-lab.github.io/aucmedi/ accessed on 10 January 2023) [20] was used to instruct a deep neural network to differentiate between two patient outcomes: survival and death.

### 2.6. Architecture

DenseNet121, a derivative of the Dense Convolutional Network (DenseNet), was implemented [21]. This architecture was selected after trying other CNNs (DenseNet201, DenseNet161, VGG19 and ResNext) because of its higher efficiency in terms of CPU requirements, image size and results. DenseNet121 stands out due to its dense connectivity pattern, its computational efficiency and minimal memory usage, which stems from the reutilization of features. It was selected for its proficiency in extracting intricate and hierarchical features from input images, a critical component in medical image analysis (Figure 1).

### 2.7. Activation Output

For our binary classification task, the softmax function was used as the final activation function. The softmax function converts output logits into probabilities by normalizing them into a probability distribution so the sum of the output probabilities equals 1. The class with the higher probability is chosen as the output prediction. Although softmax is often associated with multiclass classification problems, it is equally applicable to binary classification. The model predicts the class with the higher probability, so this might provide relevant insights to understand the probability of an outcome. Therefore, one key advantage of using softmax over sigmoid in binary classification is its interpretability as it offers a confidence level associated with the prediction that can be used to support a given clinical statement.

### 2.8. Class Imbalance and Loss Function

We initially calculated the class weights to be implemented by the categorical focal loss function. Class weights were computed using n_samples/(n_classes × bincount(y)), inspired by the work of King et al. [22]. Focal loss function prioritizes instances that are harder to classify and downplays simpler examples, thereby guiding the model to concentrate more on a balanced set of challenging samples. This method has demonstrated its robustness to class imbalance across different datasets and tasks, thereby enhancing the model’s performance [23].

### 2.9. Data Augmentation

In our CNN model, we employed several image augmentation techniques to increase the diversity and robustness of our training dataset. These techniques included mirroring (reflecting images across their vertical or horizontal axis), rotation (adjusting images by a certain degree around the center point), scaling (changing the size of the images) and elastic transformation (locally distorting the image by randomly displacing each pixel to simulate natural variations). As suggested by Isensee et al., augmentation techniques are highly efficient procedures to increase the potential generalization of a model as they allow for a better performance with unseen data [24].

### 2.10. Callbacks

In our model, we employed several callbacks, including EarlyStopping, ModelCheckpoint and ReduceLROnPlateau. EarlyStopping is used to halt training when a monitored metric has stopped improving, preventing overfitting and saving computational resources. ModelCheckpoint allows for the saving of the model after each epoch, ensuring the retention of the best performing model. On the other hand, ReduceLROnPlateau lowers the learning rate when a metric has ceased to improve, optimizing the model’s ability to find the global minimum and enhance the training performance. These strategies work together to mitigate overfitting and reduce unnecessary training time.

### 2.11. Transfer Learning

Transfer learning is a machine learning (ML) approach that applies a pre-trained model to a new, but related, task. This strategy bolsters learning efficiency, especially when data for the new task are limited. The model retains or “freezes” the learned weights from the prior task while fine-tuning the classification layer to the new task. After several epochs, the model is fully unfrozen for additional fine-tuning, thereby conserving computational resources and training time. Transfer learning was conducted for 10 epochs using the Adam optimizer with an initial learning rate of 1 × 10^−4^ and a batch size of 4 for DenseNet121.

### 2.12. Explainable Artificial Intelligence

We employed gradient-weighted class activation mapping (Grad-CAM) for explainable artificial intelligence [25]. This technique provides visual elucidations for decisions made by CNNs. It uses the gradients of any targeted concept flowing into the final convolutional layer to generate a coarse localization map that emphasizes the crucial regions in the image for a prediction of the result. The provided heat map offers insights into interpretability and helps to identify potential dataset bias.

### 2.13. Metadata

A second CNN predictive model was developed that incorporated baseline CT scan images and admission-related clinical information as the input. The clinical data were limited to the variables available upon admission (age, sex, hypertension, WFNS grade, acute hydrocephalus, etc.) and demonstrated a statistically significant association with mortality. This model was created to determine if the addition of clinical information could enhance the performance of the image-based model.

### 2.14. Statistics

Excel (Microsoft, Redmon, WA, USA; version 16.16.4) and SPSS Statistics (IBM, Armonk, NY, USA; version 24) were implemented to run conventional statistical methods. The distribution of continuous variables was assessed using a normality test. Categorical variables were expressed as frequencies and percentages. The categorical variables were compared using chi-squared and Fisher’s exact tests. The association between mortality and the continuous variables was analyzed using a Student’s *t*-test or Wilcoxon U test. Univariate analysis was performed to study the association of clinical variables with mortality at three months. The performance of the CNN was evaluated using the metrics typically implemented in DL methods such as sensibility, specificity, accuracy, F1 score and the area under the curve (AUC) for the receiver operating characteristic (ROC) curve [26].

## 3. Results

A total of 219 patients met the inclusion criteria for the study (Figure 2). Among them, 47.5% (104/219) were males and the mean age was 58 (SD = 13.06). A perimesencephalic pattern on the initial CT scan was observed in 37 patients (16.9%). In 42 cases, the initial arteriography did not detect the presence of an aneurysm; out of these, 39 cases (17.8%) were confirmed as idiopathic SAH.

Aneurismatic SAH was reported in 180 patients, with 222 aneurysms and 36 cases (20%) of multiple aneurysms. The mean WFNS and HH on admission were, respectively, 2.5 (SD = 1.6) and 2.2 (SD = 1.6), with a mode of 2 in both cases. The mean mF scale was 3.3 (SD = 0.9) and the mode was 4. For aSAH, 91 (50.5%) patients were surgically treated, 72 (40%) were endovascularly treated and 17 (9%) were not treated due to brain death signs prior to it being possible to provide any effective treatment. In 54.6% of treated patients, the aneurysm was excluded in the first 24 h after the diagnosis. Rebleeding occurred in 15 patients and only 4 of them survived. In the sample of 219 patients, the mean stay was 24 days and the mortality rate was 28.5% (Table 1).

Among the patients with SAH, mortality was significantly superior in older individuals (61.2 vs. 56.5 years old; F = 5.12; t = 2.48; *p* = 0.014), female patients (35.6% vs. 23.1%; X^2^ = 4.14; *p* = 0.042) and patients with hypertension (40.6% vs. 21.1%; X^2^ = 9.81; *p* = 0.002), intraparenchymal hematoma (48.4% vs. 21.9%; X^2^ = 15.24; *p* < 0.001) and acute hydrocephalus (42.7% vs. 19.8%; X^2^ = 13.04; *p* < 0.001). Patients with higher grades from the modified Fisher (X^2^ = 39.9; *p* < 0.001), WFNS (X^2^ = 46.9; *p* < 0.001) and HH (X^2^ = 48.6; *p* < 0.001) scales experienced higher mortality rates (Appendix A). All these variables were included as metadata in the CNN model based on baseline CT scan images and clinical information. Other cardiovascular risk factors like diabetes, dyslipidemia or smoking were not associated with a higher risk of mortality. Subdural hematoma or seizures on admission were not associated with mortality.

The highest grades on the WFNS, HH and mF scales demonstrated a strong association with mortality. Remarkably, mF grades 3 or 4 proved to be a strong risk factor for mortality compared with mF grade 1 or 2 (odds ratio of 21.7 (*p* = 0.003; 95% confidence interval: 2.91–161.71)). The results for other variables are shown in Table 2.

CNN algorithms were developed, trained, validated and tested in this study. Among the models created, the one exclusively based on the initial CT scan demonstrated the best performance. Optimal performance was achieved during the final epoch, with the following metrics: sensitivity = 0.75 (SD = 0.025; 95% CI = 0.716–0.786); specificity = 0.75 (SD = 0.025; 95% CI = 0.716–0.786); accuracy = 0.74 (SD = 0); F1 score = 0.72 (SD = 0.025; 95% CI = 0.615–0.829); and AUC (area under the curve) = 0.82 (SD = 0). The inclusion of additional clinical metadata in the model did not significantly enhance its performance. The best F1 score obtained with the combined model was as follows: sensitivity = 0.75 (SD = 0.025; 95% CI = 0.716–0.786); specificity = 0.75 (SD = 0.025; 95% CI = 0.716–0.786); accuracy = 0.74 (SD = 0); F1 score = 0.74 (SD = 0.077; 95% CI = 0.663–0.817); and AUC = 0.80 (SD = 0). The results are presented in Table 3 and depicted in Figure 2, Figure 3 and Figure 4 (Appendix A).

## 4. Discussion

In this investigation, we retrospectively reviewed all consecutive cases of SAH patients admitted to our institution and we validated our results with an external cohort from another center. Images and data were preprocessed and used to train a CNN to predict mortality in a test cohort of patients. The results demonstrated that a CNN predictive algorithm exclusively based on the initial CT outperformed a combination of images and clinical data. The results of this image-based algorithm proved the ability of the CNN to establish solid predictions using medical images as the input. We aimed to develop an innovative, open-source classification model that could readily be tested on diverse datasets. To accomplish this, we chose to utilize a standardized framework (AUCMEDI). Our methodology included preprocessing techniques like resampling, clipping and intensity normalization to minimize potential image variability. Additionally, we employed image augmentation to mitigate the risk of overfitting and to enhance the model’s efficacy on previously unseen datasets. A transfer learning approach was adopted, leveraging the pre-trained models to provide well-established and effective weights, thereby boosting the model’s performance. Lastly, the entire pipeline was not only fully open but also comprehensively documented, ensuring its availability and ease of implementation on new datasets in the future. To the best of our knowledge, this study represents the first successful development of an image-based CNN algorithm that accurately predicts mortality in patients with SAH.

Several studies have demonstrated the ability of DL models to identify abnormalities such as hemorrhages, fractures, strokes and edemas from head CT scans [27,28]. These investigations require extensive labelled image datasets as inputs to build the model [27]. Different approaches have been used for this purpose, from classifying slices as pathologic or normal to the automatic segmentation of abnormal areas [28,29,30,31]. Using AI models to accomplish iterative and tedious tasks such as blood segmentation is a significant advancement that reduces working times and allows large samples of patients to be to processed, increasing the statistical power of the clinical investigation. However, in most of the available scientific papers regarding SAH or brain hemorrhages, the automated processes are limited to feature extractions. These features are then used with conventional statistical methods or in ML algorithms, but a fully automated pipeline capable of accurately predicting a clinical outcome from raw images is lacking for SAH. In this sense, our DL model represents a further leap forward.

Regarding SAH, efforts have also focused on aneurysm detection. Using different modalities of images (CT, DSA or MRI) and approaches (stand-alone AI or AI supporting a clinician), several reports have demonstrated the ability of AI to assist in aneurysm detection [32]. Bo et al. demonstrated the utility of a DL-based model to assist radiologists in the detection of intracranial aneurysms using AngioCT [33]. Increasing the reliability, particularly the specificity, of these automated models could allow for the future screening of intracranial aneurysms in large populations in a context in which human intervention could be relegated to supervision and the final confirmation of positive results.

Mortality and outcome predictions have classically relied on risk factors and clinical and radiological scales [34]. Advanced methods of data processing represent a great opportunity to exploit the information patients harbor early on admission. Therefore, studies have implemented ML methods to extract the best from features with known implications on the final outcome. Dengler et al. compared the performance of ML methods on outcome predictions for aSAH patients and established clinico-radiological scores [35]. The authors found that GCS and age were the most relevant features for outcome predictions and that ML methods were not superior to conventional scores [35]. In a study based on clinical features and ML methods, Toledo et al. achieved an AUC for the ROC curve of 0.85 in a decision tree built with Fisher and WFNS scales to predict functional outcomes [36]. Lo et al. used an extensive database to create an predictive algorithm for outcomes [37]. This model was based on multiple demographic and clinical variables that were used as the input for a Bayesian CNN with fuzzy logic inferences [37]. The AUC for the ROC curve was 0.85. However, many of the features that fed the algorithm were not present on admission; therefore, an early prediction of patient outcomes was not feasible [37]. Our model has the ability to make predictions without any clinical input or need for expert assessments, which has potential for automatization, generalization and applicability in primary and secondary centers referring patients to tertiary hospitals.

Although it remains challenging to speculate about the potential clinical applications of this model beyond the current level of evidence, this prognostic information complements other well-known predictive factors, aiding physicians in daily decision-making for critically ill patients. Thus, we believe that certain potential applications may emerge. These include improving communication among healthcare teams, supporting the information conveyed to families, aiding in decisions related to end-of-life care, the withdrawal of invasive treatments, the implementation of rescue therapies and assisting in determining the optimal timing of treatment. Although our approach serves as an initial step in this direction, it requires further development and validation to be decisive in such a critical, intricate and ethically sensitive subject as mortality prediction. At this juncture, prior to subjecting our model to a comprehensive validation process using larger and independent datasets, it would be reckless to regard the predictions of our model as an absolute and reliable truth and guide the clinical management of these patients based solely on their varying probabilities of survival. For instance, based on a high probability of death provided by our model, clinical decisions could be skewed, resulting in prematurely discontinuing the best available treatment for a patient and presumably leading to a self-fulfilling prophecy. Therefore, the ethical challenges involved in the personalized prognostication of life-threatening conditions like SAH, particularly in terms of interpreting and conveying the inherent uncertainty to those making decisions on behalf of patients, must be considered. In this scenario, the advent of artificial-intelligence-assisted prognostication calls for a contemporary and enduring framework [38]. Such a framework should ensure that physicians, patients and their families are provided with reassurance amidst the uncertainties surrounding the unfathomable question of life and death in critically ill patients.

A paradoxical finding of our research was the null improvement of the predictive model with the addition of clinical metadata with an otherwise proven association with mortality. It was hypothesized that the CNN would extract information from the images beyond human capacity, but we also expected that the clinical data would improve the model. The image-based model was likely able to estimate the quantity and distribution of blood as well as detect signs of brain damage such as edema and the herniation and effacement of basal cisterns. Many of these radiological signs are known factors of a poor clinical grade on admission and have previously been correlated with mortality [4,13,14]. Previous works in other areas have highlighted how the clinical information adds up to an image-based model, while other groups have demonstrated exactly the opposite [39,40,41]. These conflicting experiences might be due to differences in the targeted prediction, the architecture of the NN or even how the clinical information is introduced into the model. It is also possible that baseline CT scans harbor highly valuable information that significantly impacts the outcome. This would challenge the idea of the influence of clinical management and delayed cerebral lesions on SAH mortality and emphasize the relevance of initial damage on the final outcome.

### Limitations

In addition to the retrospective nature of the sample, which might have led to the unnoticed loss of some patients, the present study harbored some limitations. First, the predictions were based on a ground truth, which originated from the results of our practice in this case. Mortality rates, causes of death, risk factors, treatment choices and overall management may vary amongst institutions. This flaw can only be tackled if larger training cohorts from different institutions representing different management protocols are used to build the model. Efforts should be made in this regard to aim for a predictive tool that can be applied to as many healthcare contexts as possible. Second, CT scanner protocols and manufacturers might change the information the model extracts from the image and, consequently, the class assigned to a particular case. However, the methodology we implemented to harmonize CT scans was designed to minimize variability in the imaging data to improve the robustness of the training as well as the accuracy and reliability of our model. Third, it can be argued that perimesencephalic SAH and aSAH are completely different diseases with vast differences in clinical evolution and outcomes; therefore, they should not be mixed in a mortality prediction study. Trained specialists in neurovascular emergencies might correctly identify a SAH as perimesencephalic with a rapid view of the CT scan and short assessment of the patient. However, one of the main applications of an image-based model such as the one herein presented is to support clinicians with their decisions, especially in non-tertiary centers where knowledge about alarm signs and prognosis might be scarce. Fourth, although the size of the training cohort was deemed to be sufficient for the construction of a precise predictive model, we acknowledge that larger samples are often preferred. As previously stated, we implemented various image augmentation techniques to increase the diversity and robustness of our training dataset. These techniques are highly effective in mitigating issues associated with smaller sample sizes and enhancing the generalizability of a model [24]. Finally, one of the main problems of DL is that the algorithm does not disclose what their decisions were based on; in other words, we cannot fully explain why the model classifies a given case into a specific class. Efforts are being made to unlock the black box that DL methods often represent. These efforts are referred as Explainable Artificial Intelligence or XAI. In DL, XAI methods are mainly post hoc, meaning that the trained model is analyzed to find learned associations [42]. Our team is currently working on visual activation maps or saliency maps based on gradient-weighted class activation mapping (Grad-CAM), which are graphic representations of the areas of an image that are important for the model to make a decision or classify a case into a group [25,43] (Figure 5 and Figure 6). The use of saliency maps will assuredly contribute to improving our understanding of prognostic model’s performance and aid in thoroughly examining each misclassified case. In this sense, saliency maps are poised to contribute to the development of future research lines. For example, if we can establish that the maps of most survivors share common patterns, this may provide valuable insights into the understanding of factors impacting the patient’s status on admission. Conversely, the maps of non-survivors may exhibit specific patterns on the initial CT scan, which could signify whether there is a critical sign demanding urgent attention or an ominous sign that would potentially render all our efforts futile.

Future research will seek to further validate the present algorithm and apply it to classification tasks such as the differentiation of perimesencephalic SAH from aSAH and the prediction of complication occurrence (vasospasm, shunt-dependent hydrocephalus, delayed cerebral ischemia, etc.).

## 5. Conclusions

DL algorithms based on initial CT scans allowed us to provide accurate predictions of mortality for SAH patients. The limited improvement seen with the addition of clinical information suggested that many factors influencing patient outcomes are present in the early stages of the disease and could be identified from the initial CT scan. AI predictive models are a promising tool that could significantly improve the understanding of, and decision-making process in, complex pathologies like SAH. However, further optimization of these models through the inclusion of more data and patients is necessary to enhance their performance on complex tasks that are beyond the potential of conventional clinical knowledge.

## Figures and Tables

**Figure 1 brainsci-14-00010-f001:**
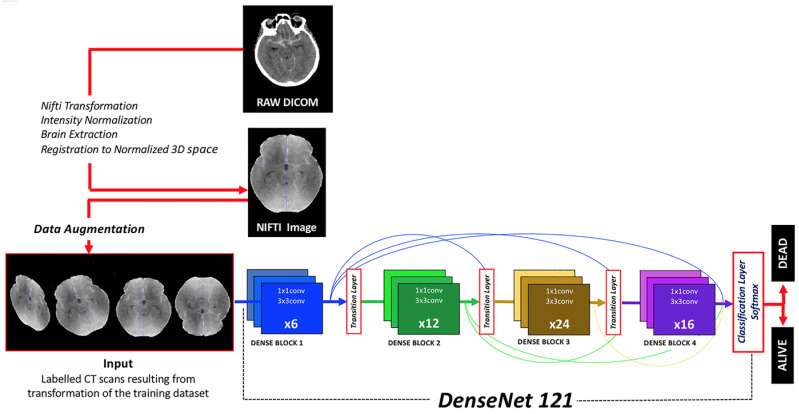
Diagram depicting the workflow from DICOM raw images to classification results provided by the algorithm. Image preprocessing, data augmentation, neural network architecture and output classification function are herein represented.

**Figure 2 brainsci-14-00010-f002:**
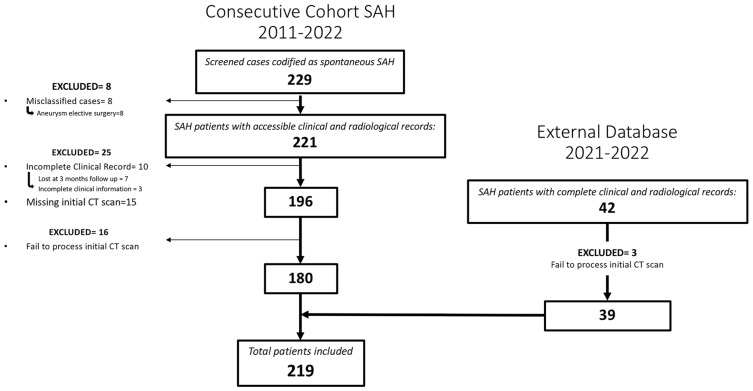
Flowchart describing the screened, included and excluded patients for the institutional and external cohort of patients whose images and data were used to create, train and evaluate the model.

**Figure 3 brainsci-14-00010-f003:**
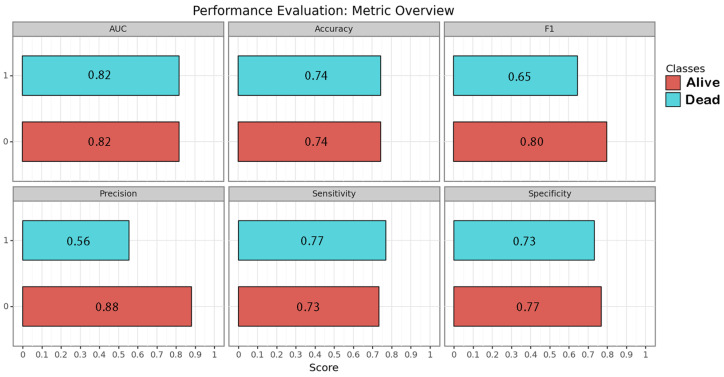
Performance of the image-based neural network algorithm. Each metric is represented for each class by its correspondent bar and numeric value within.

**Figure 4 brainsci-14-00010-f004:**
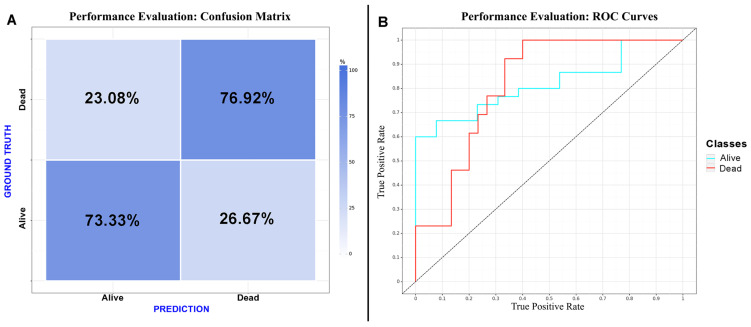
Performance evaluation: (**A**) confusion matrix of the CNN considering “dead” as a positive result for the test; (**B**) receiver operating characteristic curve of the CNN.

**Figure 5 brainsci-14-00010-f005:**
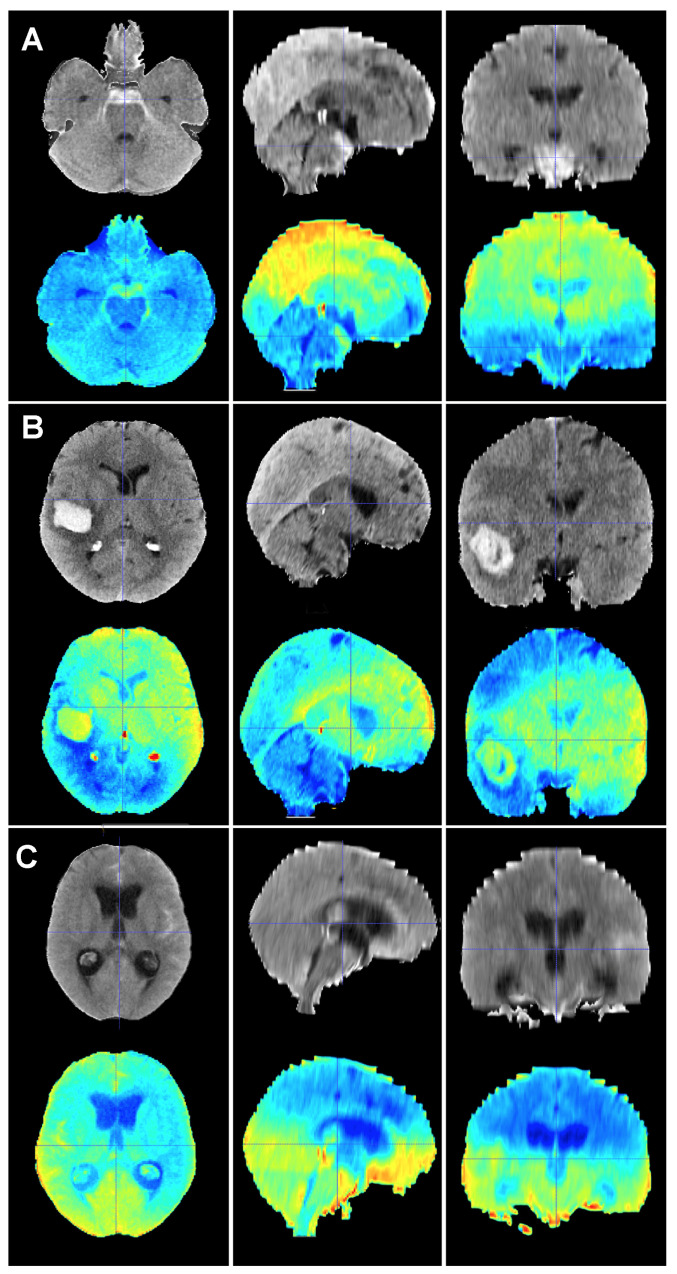
Baseline CT scan (upper row) and gradient-weighted class activation mapping or Grad-CAM (lower row) for three patients (**A**–**C**) from the test cohort who were alive three months after suffering a subarachnoid hemorrhage. Saliency maps highlighted regions in red that were more significant when classifying patients; in this case, into the group of patients who survived the event. Thus, it was possible to create a visual depiction of the process the model followed to allocate patients into each class. These maps highlighted supratentorial brain areas and seemed to disregard hemorrhages, except when they followed a perimesencephalic pattern. (**A**) A 60-year-old male who suffered a perimesencephalic SAH whose angio-MR and angiography were negative. (**B**) A 43-year-old male who was diagnosed with a SAH caused by a right middle cerebral artery aneurysm. He was admitted in good condition (WFNS 1) and was surgically treated and discharged without major neurological deficits on postoperative day 19. (**C**) A 75-year-old female who suffered a SAH and was admitted to the hospital with a WFNS grade 4. The left posteroinferior cerebellar artery aneurysm was coiled. The patient survived the event, but was still severely impaired at the three-month follow-up (modified Rankin Scale: 4).

**Figure 6 brainsci-14-00010-f006:**
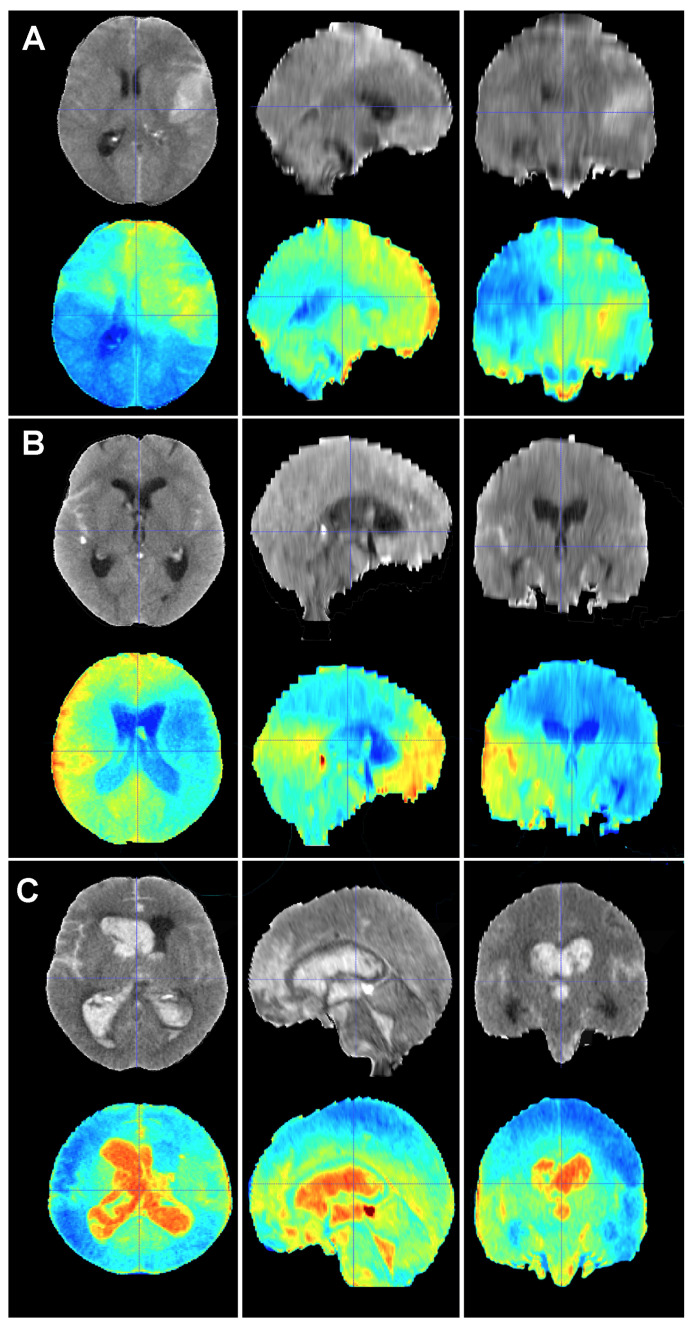
Baseline CT scan (upper row) and gradient-weighted class activation mapping or Grad-CAM (lower row) for three patients (**A**–**C**) from the test cohort who died as a result of a subarachnoid hemorrhage (SAH). These maps visually illustrate the areas the model considered to allocate patients into the “dead” group. Grad-CAM maps show that posterior fossa and intraventricular and cisternal blood might be relevant areas or items to consider in order to classify patients as dead. (**A**) A 51-year-old male who suffered a SAH due to the rupture of a left middle cerebral artery aneurysm. The patient initially presented with a WFNS grade 2, but abruptly deteriorated to a WFNS grade 5 requiring emergent surgical treatment. The patient died on postoperative day 56 as a consequence of both systemic and neurological complications. (**B**) A 70-year-old female who was diagnosed with a SAH caused by a right posterior communicating artery aneurysm who died 40 days after her admission due to a combination of factors, including delayed cerebral ischemia, meningitis and pneumonia. (**C**) A 78-year-old male with a SAH caused by an anterior communicating artery who was admitted to the hospital with a WFNS grade 5 and an mF grade 4 who died the next day after the event.

**Table 1 brainsci-14-00010-t001:** Characterization of the sample according to analyzed variables.

Variable	Mean/Mode	Number	Percentage
Female		115	52.5%
Age	57.9 (SD = 13.06) years		
RISK FACTORS			
HT		96	43.8%
Tobacco		90	41%
Smoker, Female		40	34.8% of females
HT + Tobacco		36	16.5% of total
Diabetes		20	9%
Dyslipidemia		81	37%
Familial History		3	1.5%
Idiopathic SAH		39	18%
Aneurysmal SAH		180	82%
Multiple		36	20%
Anterior Circulation		174	97%
Aneurysm Diameter	7.9 mm (SD = 5.6)		
TREATMENT			
Surgical		91	50.5%
Endovascular		72	40%
No Treatment		17	9%
Timing of Treatment			
Ultra Early (<24 h)		114	70%
Early (24–72 h)		33	20%
Delayed (>72 h)		16	10%
ADMISSION			
Hunt and Hess	2.2/2		
I		103	47%
II		45	20.5%
III		10	4.5%
IV		17	8%
V		44	20%
WFNS	2.5/2		
I		93	42.5%
II		46	21%
III		8	3.5%
IV		26	12%
V		46	21%
Modified Fisher	3.2/4		
I		15	7%
II		25	11.5%
III		35	16%
IV		144	65.5%
Intraparenchymal Hematoma		63	35%
Subdural Hematoma		9	5%
COMPLICATIONS			
Acute Hydrocephalus		96	44%
Shunt-Dependent Hydrocephalus		38	17.5%
Seizure		37	17%
Epilepsy		12	6.5%
Symptomatic Vasospasm		38	17.5%
Delayed Cerebral Ischemia		52	23.5%
Length of Stay	24 days		
OUTCOME			
mRS at 3 Months	3/6		
0		46	21%
1		37	17%
2		16	7.5%
3		19	8.5%
4		15	7%
5		23	10.5%
6		63	28.5%
Mortality		63	28.5%

HT: hypertension; mRS: modified Rankin Scale; WFNS: World Federation of Neurological Societies.

**Table 2 brainsci-14-00010-t002:** Odds ratios for variables demonstrating statistically significant association with mortality.

Variable	Reference	Degrees of Freedom	*p*-Value	OR	95% CI
Sex (Male)	Male	1	0.042	0.54	0.30–0.98
Age	1		0.014	1.03	1.01–1.05
Hypertension	Yes	1	0.002	2.55	1.41–4.63
Intraparenchimatous Hematoma	Yes	1	<0.001	3.34	1.79–6.22
Acute Hydrocephalus	Yes	1	<0.001	3.01	1.64–5.55
WFNS	1	4	<0.001		
WFNS 2			0.007	3.67	1.44–9.42
WFNS 3			0.033	5.60	1.14–27.40
WFNS 4			0.001	5.83	2.05–16.62
WFNS 5			<0.001	17.50	6.99–43.77
Hunt and Hess	1	4	<0.001		
HH 2			0.002	4.51	1.72–11.86
HH 3			0.047	5.00	1.02–24.51
HH 4			0.02	4.63	1.27–16.87
HH 5			<0.001	25.00	8.99–69.56
Modified Fisher, Dichotomized	<2	1	<0.001		
mF > 2			0.003	21.70	2.91–161.72

**Table 3 brainsci-14-00010-t003:** Performance of neural networks algorithms. Results are presented as the average of both analyzed classes (dead or alive at three-month follow-up).

Model	Image-Based Neural Network Performance				IMAGE- and Metadata-Based Neural Network Performance			
Epoch	Best AUC	Best F1	Best Loss	Last	Best AUC	Best F1	Best Loss	Last
Metric								
TP	14.5	15	15	16	16	16.5	15.5	15
TN	14.5	15	15	16	16	16.5	15.5	15
FP	7	6.5	6.5	5.5	5.5	5	6	6.5
FN	7	6.5	6.5	5.5	5.5	5	6	6.5
Sensitivity	0.53	0.61	0.74	0.75	0.60	0.75	0.69	0.50
Specificity	0.53	0.61	0.74	0.75	0.60	0.75	0.69	0.50
Precision	0.56	0.63	0.70	0.72	0.75	0.73	0.68	0.35
FP Rate	0.47	0.39	0.26	0.25	0.40	0.25	0.31	0.50
FN Rate	0.47	0.39	0.26	0.25	0.40	0.25	0.31	0.50
FDR	0.44	0.37	0.30	0.28	0.25	0.27	0.32	0.15
Accuracy	0.67	0.70	0.70	0.74	0.74	0.77	0.72	0.70
F1	0.51	0.61	0.69	0.72	0.60	0.74	0.68	0.41
AUC	0.72	0.74	0.73	0.82	0.78	0.80	0.78	0.35

FDR: false discovery rate; FN: false negative; FP: false positive; TN: true negative; TP: true positive.

## Data Availability

The source code for the AUCMEDI framework can be found at https://github.com/frankkramer-lab/aucmedi. Additionally, the pipeline utilized in our study is publicly available at https://github.com/smcch/Subarachnoid_Hemorrhage_segmentation_and_mortality_prediction. These repositories provide access to the respective source codes, enabling researchers and interested individuals to explore and utilize the frameworks and pipelines implemented in our study.

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
