# Peer review of "Mortality Prediction of Patients with Subarachnoid Hemorrhage Using a Deep Learning Model Based on an Initial Brain CT Scan"

_brainsci, 2023, doi:10.3390/brainsci14010010_

Round 1

Reviewer 1 Report

Comments and Suggestions for Authors

Dear Authors,

I am glad to have the opportunity to review your work. The aim of the paper was to predict mortality in SAH patients by processing the initial CT scan on a CNN based algorithm

 The topic has interest for readers, it is well written and structured and with interesting results. Study sample is small, which is the main shortcoming.

In Methods, please state clearly inclusion criteria. Also, when you say that patients from other center were included…you mention similar characteristics. Please state inclusion criteria for all of the patients. Also, exclusion criteria. Also, add study type according to STROBE. Mino grammar corrections are needed.

I suggest minor revision of the paper.

Comments on the Quality of English Language

Minor correction is needed.

Author Response

Dear Reviewer,

Thank you very much for taking the time to review this manuscript. Please find the detailed responses below. The corresponding corrections are highlighted in the resubmitted files.  

-Inclusion and exclusion criteria for both cohorts have been clearly disclosed in methods section.

-Information regarding study type (retrospective non-interventional) has been added.

-English grammar and spelling revision has been conducted. Changes can be tracked on the latest submitted file.

Thank you for your contribution, your recommendations are highly appreciated.

Reviewer 2 Report

Comments and Suggestions for Authors

The paper requires major revisions. As a reviewer, I have several questions and comments regarding your paper. I believe addressing these concerns will help improve the clarity and completeness of your research:

1. Could you provide more details on the selection criteria for the patients included in the study? Specifically, how were patients with aneurysmal SAH distinguished from those with non-aneurysmal SAH?

2. Regarding the neural network architecture, why was DenseNet-121 chosen for this study? Were there specific features of DenseNet-121 that made it suitable for predicting mortality in SAH patients based on CT scan images?

3. In the activation output section, you mentioned the use of the softmax function. Could you elaborate on why this specific activation function was chosen for the binary classification task of predicting survival and death?

4. Regarding class imbalance and loss function, how were class weights calculated, and why was the categorical focal loss function selected for implementation?

5. In the data augmentation section, you mentioned employing several image augmentation techniques. Could you provide more details on the specific augmentation techniques used and the rationale behind choosing them?

6. What were the specific criteria used for selecting Callbacks such as EarlyStopping, ModelCheckpoint, and ReduceLROnPlateau in the neural network model?

7. Please elaborate on the transfer learning approach applied in this study. How was the pre-trained model adapted to the specific task of predicting mortality in SAH patients?

8. In the Explainable Artificial Intelligence section, you mentioned the use of Gradient-weighted Class Activation Mapping (Grad-CAM). Could you explain how this technique was employed to provide visual elucidations for decisions made by the CNN?

9. Could you provide more information on the clinical variables that were included in the second CNN predictive model? How were these variables selected, and what impact did their addition have on the model's performance?

10. In the statistics section, you mentioned using Sensibility, Specificity, Accuracy, F1 score, and AUC for evaluating the performance of the CNN. Could you discuss the rationale behind choosing these specific metrics and their relevance in the context of predicting mortality in SAH patients?

11. What steps were taken to ensure the robustness and reproducibility of the study, especially in terms of the image preprocessing and neural network training?

12. Were there any limitations or challenges encountered during the image preprocessing phase, and how were they addressed?

13. How were missing data handled in the study, particularly in the clinical records of patients?

14. In the Results section, you provided demographic information about the patient sample. Were there any notable demographic characteristics that may have influenced the study outcomes, and were these factors controlled for in the analysis?

15. Could you discuss any potential biases that may have affected the study results and how they were mitigated?

16-Could you provide more information about the clinical implications and potential applications of your findings? How might this research influence clinical practice or patient outcomes? Add suitable references to the site with [ PMID: PMID: 37701174]

17. The study mentions that the model showed good accuracy at predicting mortality. Can you provide additional insights into the false positives and false negatives, and any implications of these results in a clinical setting?

18. Do you have any recommendations for future research directions based on the findings and limitations of this study?

Reviewer 3 Report

Comments and Suggestions for Authors

This paper reports on a 2-center retrospective cohort study using convolutional neural networks and deep learning of the ability of CT brain scan taken at baseline after subarachnoid haemorrhage to predict mortality 3 months. The authors used patients from their centre for the training and those from another centre as a validation cohort. They found an accuracy of 74% and AUC of 82%.

The authors may wish to attend to the following issues:

1.       Title - ‘enhanced’ - compared to ??, as only CT images are used, as opposed to comparing clinical vs clinical + images? Maybe this word can be deleted? To add ‘brain’ to CT .. scan’

2.      Line 17 - to spell out ‘AUCMEDI” at first use

3.      Line ‘33’ – ‘marked variations in specific regions’ – do the authors mean there are variations within each of these regions?

4.      Lines 38-39 - ‘little change’ and 48 ‘scarcely improved’ – this may not be true - I draw the attention of the authors to these 2 papers 1: Lovelock CE, Rinkel GJ, Rothwell PM. Time trends in outcome of subarachnoid hemorrhage: Population-based study and systematic review. Neurology. 2010 May 11;74(19):1494-501. doi: 10.1212/WNL.0b013e3181dd42b3. Epub 2010 Apr 7. PMID: 20375310; PMCID: PMC2875923. AND Nieuwkamp DJ, Setz LE, Algra A, Linn FH, de Rooij NK, Rinkel GJ. Changes in case fatality of aneurysmal subarachnoid haemorrhage over time, according to age, sex, and region: a meta-analysis. Lancet Neurol. 2009 Jul;8(7):635-42. doi:10.1016/S1474-4422(09)70126-7. Epub 2009 Jun 6. PMID: 19501022.

5.      Line 50 - ‘challenge’ – I refer to authors to this paper - de Winkel J, Cras TY, Dammers R, van Doormaal PJ, van der Jagt M, Dippel DWJ, Lingsma HF, Roozenbeek B. Early predictors of functional outcome in poor-grade aneurysmal subarachnoid hemorrhage: a systematic review and meta-analysis. BMC Neurol. 2022 Jun 30;22(1):239. doi: 10.1186/s12883-022-02734-x. PMID: 35773634; PMCID: PMC9245240.

6.      Line 64 - ‘single sequence… to predict a patient's course’ – this does seem a bit simplistic?

7.      Table 1 – ‘Dyslipidemia’ is mis-spelt

8.      Fig 2 – please provide data on the TOTAL number of SAH patients admitted to EACH center during the study period from which the 221 subjects where derived and tested upon, and the reasons for the exclusion of the others

9.      (I have no access to the supplementary data)

10.  There is no visible data on the number/proportion lost to follow-up at 3 months and how these missing values were managed statistically

11.  Line 304 – additional limitations include single centre patients for training, small numbers, no data on apriori excluded patients, etc

12.  I agree with the authors that the data may only apply at best to the 2 centres involved in the study. No data was provided on how the patients were managed other than specifically for the aneurysm eg use of nimodipine, admission blood pressure, glucose control, rehabilitation, etc

13.  The names of the 2 centers should be clearly mentioned in the paper

Round 2

Reviewer 2 Report

Comments and Suggestions for Authors

I would like to enhance the article by addressing specific questions, comments, and references. However, it is unclear where these were addressed, and I require precise answers regarding their locations. Questions 5, 6, 10, 16, and 17 have not been answered accurately and correctly.

Author Response

  1. In the data augmentation section, you mentioned employing several image augmentation techniques. Could you provide more details on the specific augmentation techniques used and the rationale behind choosing them?

The specific techniques of data augmentation employed in our pipeline are summarised in the manuscript. We reference to the article by Isensee et al. in which it is explained and illustrated how avoiding data augmentation steps results in worse performances in segmentation tasks.

Additional information has been added to the text: “In our CNN model, we employed several image augmentation techniques to increase the diversity and robustness of our training dataset. These techniques included mirroring (reflecting images across their vertical or horizontal axis), rotation (adjusting images by a certain degree around the center point), scaling (changing the size of the images), and elastic transformation (distorting the image locally by displacing each pixel randomly, which simulates natural variations). As suggested by Isensee et al. augmentation techniques are highly efficient procedures to increase the potential generalization of the model as they allow for a better performance on unseen data[24]."

  1. What were the specific criteria used for selecting Callbacks such as EarlyStopping, ModelCheckpoint, and ReduceLROnPlateau in the neural network model?

These callbacks help in managing the training process efficiently, avoiding overfitting, and ensuring that the best possible model is saved.

EarlyStopping:  To prevent overfitting by stopping the training when a monitored metric (like validation loss) stops improving.

ModelCheckpoint: To save the model at certain intervals or when it surpasses previous performance benchmarks.

ReduceLROnPlateau: To reduce the learning rate when a metric has stopped improving, helping to fine-tune the model.

Information on this regard has been added to the text: In our model, we employed several Callbacks including EarlyStopping, ModelCheckpoint, and ReduceLROnPlateau. EarlyStopping is used to halt training when a monitored metric has stopped improving, preventing overfitting and saving computational resources. ModelCheckpoint allows for the saving of the model after each epoch, ensuring the retention of the best performing model. ReduceLROnPlateau, on the other hand, lowers the learning rate when a metric has ceased to improve, optimizing the model's ability to find the global minimum and enhance the training performance. These strategies work together to mitigate overfitting and reduce unnecessary training time.

  1. In the statistics section, you mentioned using Sensibility, Specificity, Accuracy, F1 score, and AUC for evaluating the performance of the CNN. Could you discuss the rationale behind choosing these specific metrics and their relevance in the context of predicting mortality in SAH patients?

Discussing what could be preferred in a prognostic model for mortality prediction in SAH patients might be cumbersome. Specially when non aneurismatic and aneurysmatic SAH are included. Higher Mortality prediction accuracy (sensibility) might lead to nihilistic decissions which would be very harmful for salvageable patients. We deliberately avoided to elaborate on this, because we consider that this model requires further validation before discussing its ethical implications.  Information was added to the text and a reference to the work by Muller et al. is included :

Selected model’s performance metrics are the gold-standard in the field of medical image analysis for classification and segmentation tasks.

16-Could you provide more information about the clinical implications and potential applications of your findings? How might this research influence clinical practice or patient outcomes? Add suitable references to the site with [ PMID: PMID: 37701174]

It is important to emphasize that this algorithm represents an initial effort that will require further validation before its application in a real clinical context. Nevertheless, the type of information that a classification model with prognostic intent can provide would potentially be useful in informing patients and their families, in deciding the timing of therapeutic interventions, in studying the factors that influence a patient's progression, in the aggressiveness of salvage therapies, etc. Nonetheless, the arising ethical concerns of AI based prognostic information should be subject of in-depth consideration and cautious conclusions.

 Information and bibliographic references have been added to the manuscript.

Nonetheless, the ethical challenges involved in personalized prognostication of life-threatening conditions like SAH, particularly in terms of interpreting and conveying the inherent uncertainty to those making decisions on behalf of patients, must be considered. In this scenario, the advent of artificial intelligence-assisted prognostication calls for a contemporary and enduring framework(5). Such a framework should ensure that physicians, patients, and their families are provided with reassurance amidst the uncertainties surrounding the unfathomable question of life and death in critically ill patients.

  1. The study mentions that the model showed good accuracy at predicting mortality. Can you provide additional insights into the false positives and false negatives, and any implications of these results in a clinical setting? 

 The limited number of false positives and negatives in the test cohort hindered our ability to draw solid conclusions about the reasons for the model's classification mistakes. Employing saliency maps, however, will undoubtedly contribute to improve our comprehension of the model's performance and particularly analyze each misclassified case.

Information has been added to the text:

The use of saliency maps will assuredly contribute to improve our understanding of prognostic model's performance and aid in thoroughly examining each misclassified case. 

  Reviewer 3 Report

Comments and Suggestions for Authors

This is a revised submission of a paper reporting on a 2-center retrospective cohort study using convolutional neural networks and deep learning of the ability of CT brain scan taken at baseline after subarachnoid haemorrhage to predict mortality 3 months.

The authors have attended to most of the issues:

1.     Title - To add ‘brain’ to CT scan’ was not done

2.     ‘There is no visible data on the number/proportion lost to follow-up at 3 months and how these missing values were managed statistically’ – I see from the authors reply that ‘patients without available clinical information at 3 months were excluded’ – how many were there? And shouldn’t they be included in the numbers preceding the 221 and 42? This is contra to the authors reply that ‘no SAH cases were lost’. I would expect something along the following lines – admitted to hospital A for SAH (based on hospital DRG codes at discharge) – 250; SAH patients with accessible clinical and radiological records – 221;…….. total number of patients included – 219; number with 3 month data 200. And to explain the loss at each step

Author Response

Thank you for your revision. Herein our comments and changes according to your suggestions.

1.- The tittle has been changed attending to reviewer´s suggestion.

2.- Number of patients lost in each step is explained on Fig.2.

The consecutive cohort of our hospital 2011-2022 was prospectively built in the past, with periodic reviews of admissions, surgeries and procedures performed in that frame of 11 years. There was a small amount of misclassified cases in that dataset (8), which after revision were excluded because they were not actual cases of spontaneous SAH. Those 8 cases were, actually, aneurysm elective cases. Nonetheless we have modified the diagram to include this information and extend the information of reasons leading to case exclusion.

There is no losses in the external cohort, as it was thoroughly and prospectively collected along 1 year. The only cases lost from that dataset were due to specific issues during the preprocessing phase (3)